# Enhanced bacterial chemotaxis in confined microchannels occurs at lane widths matching circular swimming radius

Caijuan Yue[1†], Chi Zhang[1*†], Rongjing Zhang[1*], Junhua Yuan[1,2*]

[1]Hefei National Research Center for Physical Sciences at the Microscale, and Department of Physics, University of Science and Technology of China, Hefei, China; [2]College of Physics, Guizhou University, Guiyang, China

## eLife Assessment

This **important** work examines the effects of side-wall confinement on chemotaxis of swimming bacteria in a shallow microfluidic channel. The authors present **convincing** experimental evidence, combined with geometric analysis and numerical simulations of simplified models, showing that chemotaxis is enhanced when the distance between the side walls is comparable to the intrinsic radius of chiral circular swimming near open surfaces. This study should be of interest to scientists specializing in bacteria-surface interactions.

*For correspondence:
zhchi@ustc.edu.cn (CZ);
rjzhang@ustc.edu.cn (RZ);
jhyuan@ustc.edu.cn (JY)

†These authors contributed equally to this work

Competing interest: The authors declare that no competing interests exist.

**Abstract** Understanding bacterial behavior in confined environments is helpful for elucidating microbial ecology and developing strategies to manage bacterial infections. While extensive research has focused on bacterial motility on surfaces and in porous media, chemotaxis in confined spaces remains poorly understood. Here, we investigate the chemotaxis of *Escherichia coli* within microfluidic lanes under a linear concentration gradient of L-aspartate. We demonstrate that *E. coli* exhibits significantly enhanced chemotaxis in lanes with sidewalls compared to open surfaces. We attribute this phenomenon primarily to the intrinsic chiral clockwise circular motion of surface-swimming bacteria and the subsequent alignment effect upon collision with the sidewalls. By varying lane widths, we identify that an 8 μm width—approximating the radius of bacterial circular swimming on surfaces—maximizes chemotactic drift velocity. These results are supported by both experimental observations and stochastic simulations, establishing a clear proportional relationship between optimal lane width and the radius of bacterial circular swimming. Further geometric analysis provides an intuitive understanding of this phenomenon. Our results may offer insights into bacterial navigation in complex biological environments such as host tissues and biofilms, providing a preliminary step toward exploring microbial ecology in confined habitats and potential strategies for controlling bacterial infections.

## Introduction

Microorganisms inhabit a variety of complex confined environments. Sperm swimming in the reproductive tract of a female animal (*Suarez and Pacey, 2006*), bacteria navigating through host cells and tissues (*Bray, 1992*), cells moving in the wrinkle of biofilm (*Geisel et al., 2022*), and swarm cells maneuvering in crowded cell clusters (*Tian et al., 2021*) are all examples of organisms operating in geometrically confined environments. Furthermore, many diseases have been found to be related to bacterial motion in confinement, such as urinary tract infections and neonatal sepsis.

Previous research has explored the motion of bacteria in confined environments. Studies have shown that flagellated bacteria tend to accumulate near both solid-liquid (*Berke et al., 2008*; *Li and Tang, 2009*) and air-liquid interfaces (*Morse et al., 2013*), and exhibit a specific chirality in their direction of circular swimming (*Lauga et al., 2006*). For example, *Escherichia coli* swim clockwise when observed from above a solid surface (*DiLuzio et al., 2005*; *Lauga et al., 2006*; *Junot et al., 2022*; *Molaei et al., 2014*), whereas *Caulobacter crescentus* move in tight, counter-clockwise circles when viewed from the liquid side. Moreover, Tang et al. found that flagellated bacteria orbit within the thin fluid film around micrometer-sized particles (*Araujo et al., 2019*). Bacteria also display different motion behaviors in porous media and tubes (*Bhattacharjee and Datta, 2019b*; *Bhattacharjee and Datta, 2019a*; *Figueroa-Morales et al., 2020*; *Licata et al., 2016*; *Shrestha et al., 2023*; *Tokárová et al., 2021*; *Wioland et al., 2016*), as well as in other types of confinement (*Chen et al., 2024*; *Mushenheim et al., 2014*; *Woolverton et al., 2005*). In narrow channels with width less than 4 µm, bacterial growth (*Männik et al., 2009*), swimming trajectories (*Lynch et al., 2022*), and swimming speeds (*Vizsnyiczai et al., 2020*) are significantly impacted compared to bulk conditions. Furthermore, studies on bacterial transport in rectangular microchannels under flow have demonstrated that confinement influences both bacterial fluxes and spatial distribution, with cells exhibiting distinct transport dynamics along channel edges (*Figueroa-Morales et al., 2015*). However, while motility in confinement is well documented, the specific rules governing bacterial chemotaxis within geometrically constrained environments remain less understood.

*E. coli* can sense and move in chemical gradients through a set of chemotactic proteins (*Hazelbauer et al., 2008*; *Sourjik, 2004*). The transmembrane chemoreceptors detect attractants or repellents and transmit signals into the cell by modulating the autophosphorylation of the histidine kinase CheA. Attractant binding suppresses CheA autophosphorylation, while repellent binding promotes it. This modulation alters the concentration of the phosphorylated response regulator protein, CheY-P (*Bren and Eisenbach, 2000*; *Falke and Hazelbauer, 2001*; *Salah Ud-Din and Roujeinikova, 2017*; *Sourjik, 2004*). CheY-P binds to the cytoplasmic domain of flagellar motors, altering the probability of their clockwise rotation (*Sarkar et al., 2010*; *Wadhams and Armitage, 2004*). Clockwise and counterclockwise rotations of motors lead to reorientation (tumble) and smooth swimming (run) modes of the cell body, respectively (*Baker et al., 2006*). The adaptation enzymes CheR and CheB methylate and demethylate the receptors, respectively, mediating sensory adaptation (*Oosawa and Imae, 1984*; *Weis et al., 1990*).

While the molecular mechanism is well characterized, the physical environment plays a crucial role in chemotactic efficiency. Bacterial chemotaxis has been observed under various degrees of confinement, often yielding complex behaviors. For instance, in quasi-2D microchannels with a depth of 2 µm, bacteria were found to adjust their run lengths and tumble frequencies to respond to chemical gradients (*Raza et al., 2023*). Similarly, studies in 10-µm-depth microchambers revealed that *E. coli* maintains chemotactic performance in the presence of evenly spaced obstacles (*Rashid et al., 2019*). While these studies focused on confinement between top and bottom surfaces, recent work by Braham et al. showed that bacteria on a single solid surface can exhibit an escape response to high-concentration chemorepellents (250 mM Ni(NO$_3$)$_2$) (*Braham et al., 2024*). Conversely, other research has shown that bacterial chemotactic drift velocity decreases sharply near sample chamber surfaces compared to bulk liquid, a phenomenon attributed to circular cell movement at these interfaces (*Grognot and Taute, 2021*). This observation suggests that a single surface can significantly impair bacterial chemotactic performance. Yet, a study of chemotaxis in two-dimensional bacterial swarms on semisolid agar surfaces has demonstrated that *E. coli* can exhibit effective chemotaxis in attractant gradients, especially at high cell densities (*Tian et al., 2021*). This apparent contradiction is likely due to cell-cell interactions mitigating surface effects in dense populations. Despite these investigations into motility under surface effects, a comprehensive understanding of how specific confinement geometries modulate chemotaxis remains limited.

Here, we study the chemotaxis of *E. coli* on surfaces within lanes of different widths, exposed to a linear concentration gradient of L-aspartate using a hydrogel-based microfluidic device. Our findings reveal enhanced chemotactic performance in the presence of sidewalls compared to their absence. Cells in the middle area (MA) exhibited no drift and did not contribute to overall chemotaxis. However, cells near the left sidewall (LSW) and right sidewall (RSW) demonstrated negative and positive drift, respectively, with the positive drift velocity from the RSW exceeding the negative drift velocity from the LSW.

Furthermore, by comparing lanes of different widths within the same stable linear gradient field, we observed that cells exhibited the highest drift velocity in 8-μm-wide lanes, which approximates the peak value in the distribution of circular swimming radii on surfaces. This optimal width for maximal chemotaxis resulted from the largest percentage of cells occupying the RSW. Stochastic simulations of bacterial chemotaxis corroborated that the optimized lane width varies with the bacteria's circular swimming radius. Our findings have important implications for the screening of swimming organisms and the study of chemotactic behavior in real-world scenarios.

## Results

### Chemotaxis of *E. coli* within lane confinements

To study bacterial chemotaxis in confined environments such as tubes or interstitial tissues, we developed a hydrogel-based microfluidic device featuring lanes of different widths. The device design, illustrated in *Figure 1A*, incorporates two 2% (wt/vol) agarose walls inserted between three main channels: a sink channel, a cell channel, and a source channel. These walls prevent cell passage while allowing chemical diffusion. The agarose walls and main channels measure 100 μm and 400 μm in width, respectively, with the device having a depth of 30 μm.

We established a linear concentration gradient (0.05 μM/μm) of L-aspartate, an attractant sensed by the Tar receptor, by flowing motility medium through the sink channel and 50 μM L-aspartate solution (prepared in motility medium) through the source channel at a constant rate of 10 μl/min. The resulting gradient across the cell channel is depicted in *Figure 1—figure supplement 1*.

The lanes of various widths were created using parallel PDMS (polydimethylsiloxane) pillars spaced at different distances. Each lane, approximately 160 μm in length, is highlighted by a yellow rectangular box in *Figure 1A*. *E. coli* cells, confined to the cell channel, sensed the attractant gradient and performed chemotaxis within these lanes of different widths.

To enhance imaging contrast, we introduced a plasmid expressing mCherry fluorescent protein (pTrc99a-mCherry) into wild-type *E. coli* HCB1, denoting it as HCB1-pTrc99a-mcherry. A high-sensitivity CMOS camera was used to record bacterial movement trajectories on the bottom surfaces of the lanes in the cell channel, where most cells accumulate due to the combined effects of gravity and hydrodynamic wall attraction in the Percoll-free medium.

We recorded the movement of HCB1-pTrc99a-mCherry cells on the liquid-glass surface of a 44-μm-wide lane with and without an L-aspartate concentration gradient, respectively. An example video is shown in *Video 1*, and several representative trajectories are illustrated in *Figure 1B*.

Considering the potential escape into the third dimension at the edge (*Figueroa-Morales et al., 2015*), we calculated the drift velocities $v_d$ of the trajectories from both assays along the x-direction (up-gradient) by fitting the relation $\langle x(i+n) - x(i) \rangle \sim n\Delta t$ (n=1, 3, 5, …, 39) with a linear function, the slope of which is the drift velocity. Here, $x(i)$ represents the cell's x-position in the ith frame, and $\Delta t$ denotes the time interval between two frames (0.05 s). This method yields a drift velocity that is independent of potential cell loss into the third dimension. As shown in *Figure 1C*, the drift velocity $v_d$ was 1.6±0.3 μm/s and 0.2±0.2 μm/s (mean±SD) for the gradient and control (no-gradient) assays, respectively. Thus, bacteria in the L-aspartate gradient exhibited obvious chemotaxis on the lane surfaces, contrasting with previous findings on liquid-solid surfaces without sidewalls (*Grognot and Taute, 2021*).

### The cells in the RSW region dominated the chemotaxis of *E. coli* within lane confinements

To elucidate the mechanism of bacterial chemotaxis within lane confinements, we analyzed cell trajectories, identifying three distinct motion states due to the presence of sidewalls. As illustrated in *Figure 2A*, cells exhibited straight-line swimming along sidewalls, movement toward or away from sidewalls, or circular swimming. *Figure 2B* displays three representative trajectories. Circular swimming is characteristic of typical *E. coli* surface motion (*Lauga et al., 2006*). We further analyzed the trajectories of bacteria colliding with the sidewalls. As shown in *Figure 2—figure supplement 1*, cells colliding with the sidewalls tend to align with and swim along the sidewalls, and the alignment timescale is negligible compared to the typical wall residence time (*Li and Tang, 2009*).

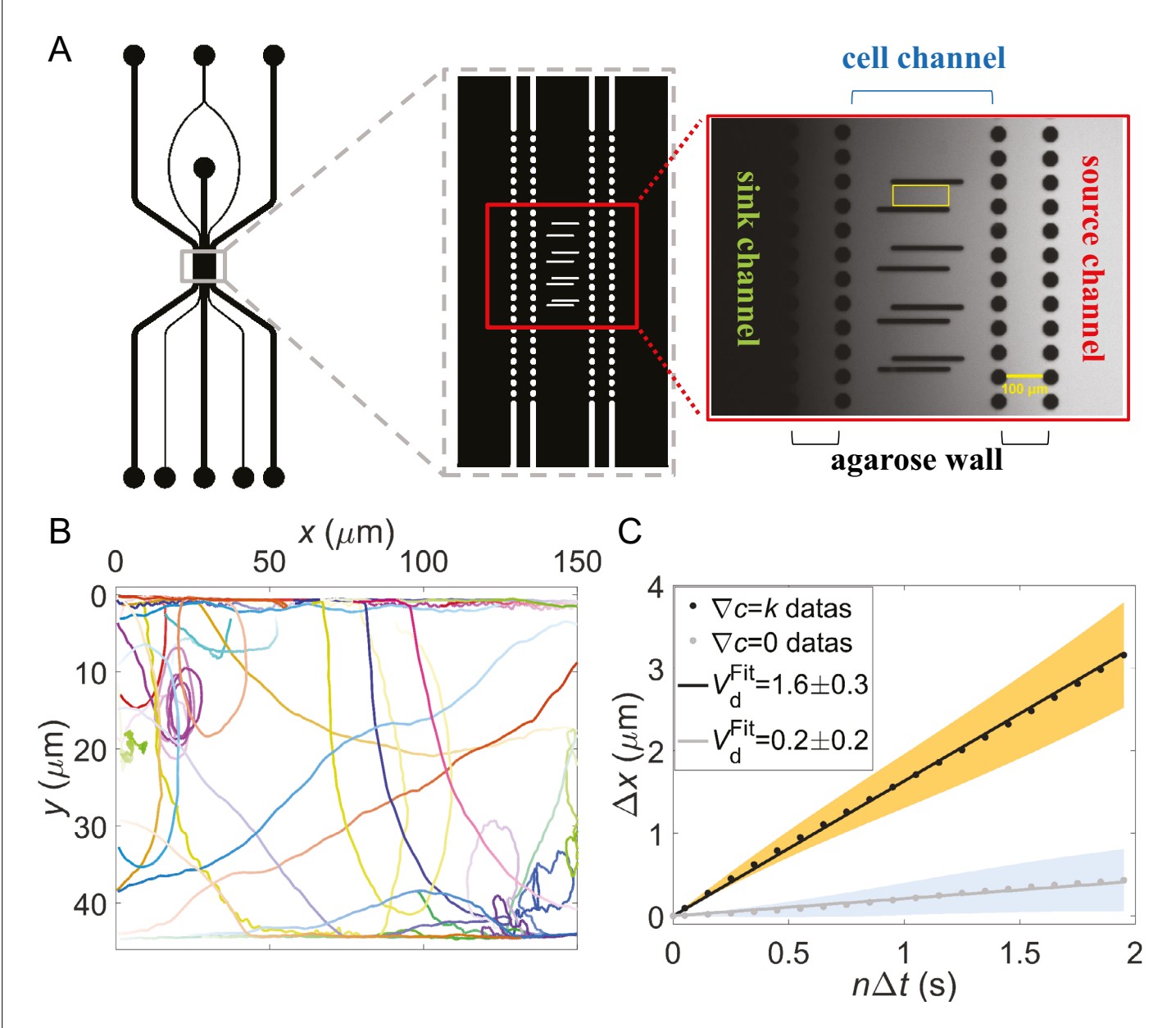

**Figure 1.** Chemotaxis of *E. coli* in confined lanes. (**A**) Microfluidic device used in the experiment (top view). (**B**) Thirty bacterial trajectories selected from the data of 44-μm-wide lane in gradient assays. Distinct colors denote individual trajectories, with color intensity darkening to indicate time progression. These represent a subset of the trajectories analyzed in panel (**C**). (**C**) The relationship between $\Delta x$ and $n\Delta t$ calculated from all trajectories in the lanes with a width of 44 μm. The black dots represent the gradient assay ($\nabla c = 0.05$ μM/μm) with a total of 3206 tracks from 6 movies, while the gray dots represent no gradient ($\nabla c = 0$) with a total of 4755 tracks from 10 movies. The light-yellow and light-blue shadows represent the standard error of the mean (SEM) of different trajectories in the gradient assays and the control assays, respectively. Linear fitting was performed to obtain $V_d = 1.6 \pm 0.3$ μm/s (black solid line) and $V_d = 0.2 \pm 0.2$ μm/s (gray solid line) for gradient and control assays, respectively. Error in drift velocity represents standard deviation.

The online version of this article includes the following figure supplement(s) for figure 1:

**Figure supplement 1.** Gradient calibration of microfluidics with fluorescein.

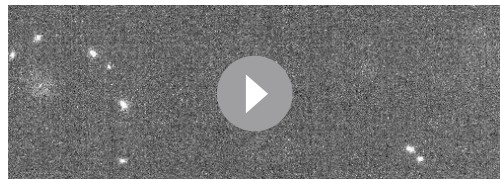

**Video 1.** An example video of HCB1-pTrc99a-mCherry cells chemotaxis in a 44-µm-wide lane under a linear gradient of L-aspartate. The source channel is on the right side.

https://elifesciences.org/articles/102686/figures#video1

We distinguished these motion states by calculating the rotational exponent ($\gamma_R$) for each trajectory, fitting the mean-squared orientational displacement <MSOD> for $\tau$<0.4 s:

$$\langle MSOD \rangle = \left\langle \left( \theta\left(t+\tau\right) - \theta\left(t\right) \right)^2 \right\rangle = C_R \tau^{\gamma_R},$$

where $\theta\left(t\right)$ is the angle between the swimming direction and the positive *x*-axis direction at time *t*, $\tau$ is the lag time, $C_R$ is a constant, and $\gamma_R$ is the rotational exponent. *Figure 2C* shows the $\gamma_R$ values for the three trajectories in *Figure 2B*, while *Figure 2D* illustrates the relationship between $\gamma_R$ and average *y*-coordinate. Trajectories further from sidewalls exhibit larger $\gamma_R$, indicating more curved paths, while $\gamma_R$ decreases sharply as the cells approach sidewalls, representing straighter trajectories.

We also analyzed tumble kinematics for cells swimming along and away from sidewalls. *Figure 2E and F* shows normalized tumble angle distributions for cells in sidewalls (SW) and MA, respectively. Red solid lines represent exponential function fits $y=a* \exp\left(-x/b\right)$, with fitted *b* values of 0.25 rad (SW) and 1.35 rad (MA). Consistent with previous reports (*Lemelle et al., 2020*), we observed a higher occurrence of large reorientation angles for surface-swimming cells in the MA region (*Figure 2F*). Interestingly, we also found large tumble angles in the SW region, albeit with a smaller characteristic angle than in the MA region. *Video 2* demonstrates an example of a large reorientation angle during tumbling.

Given the strong spatial dependence of trajectory characteristics, we classified the cells into three distinct populations. The distribution of cells along the *y*-axis is shown in *Figure 2G*. Based on mean distances from the sidewalls, we divided the lane into three areas (viewing along the positive *x*-axis): the LSW, the MA, and the RSW. As shown by different shaded colors in *Figure 2D and G*, we defined cells in LSW, RSW, and MA for trajectories with $\langle y \rangle \leq d$, $\langle y \rangle \geq w-d$, and $d< \langle y \rangle <w - d$, respectively. The threshold *d* was set at 3 µm (approximately the average length of a cell body), and *w* represents the lane width. The proportion of trajectories in different regions is shown in *Figure 2I*. For the gradient assay, the probabilities of trajectories being in LSW, MA, and RSW regions were 0.20±0.01, 0.59±0.02, and 0.21±0.03, respectively. In the control assay without gradient, these values were 0.18±0.03, 0.70±0.03, and 0.12±0.01, respectively.

We then compared the drift velocity of cells in the three areas separately for both gradient and control assays. As shown in *Figure 2H*, cells in the LSW region tended to display a negative drift with $v_d^{LSW}$ = –5.7 ± 2.1 µm/s and –5.0±2.0 µm/s for gradient and control assays, respectively. Cells in the MA region exhibited minimal drift with $v_d^{MA}$ = –0.3 ± 1.2 µm/s and 0.4±0.6 µm/s for gradient and control assays, respectively. Cells in the RSW region displayed a typical positive drift with $v_d^{RSW}$ = 14.8 ± 1.2 µm/s and 6.2±1.9 µm/s for gradient and control assays, respectively. Thus, cells in the RSW region dominated the difference in drift velocity between the two assays.

In summary, cells swim on the right-hand side under the effect of the bottom surface of lanes (*DiLuzio et al., 2005*), a behavior fundamentally dictated by the chirality of their surface interaction. The LSW and RSW cells swim down and up the gradient, respectively, while MA cells swim in circles without any drift. RSW cells demonstrated more persistent swimming toward the attractant in the L-aspartate gradient, exhibiting a higher positive drift velocity. To further illustrate the significance of this chirality, consider a hypothetical scenario where the bacteria exhibited counterclockwise circular swimming on the surface. In such a case, the hydrodynamic interaction would cause cells to veer to the left rather than the right. Consequently, the cells swimming up-gradient would accumulate along LSW, while those swimming down-gradient would accumulate along RSW. While the overall enhancement of chemotaxis via confinement would persist, the spatial distribution of the up- and down-gradient populations would be mirrored relative to the channel axis. Detailed information on the difference in drift velocity between LSW and RSW cells can be found in Appendix 1. Notably, the attractant gradient minimally affects the ratio of bacteria in different regions.

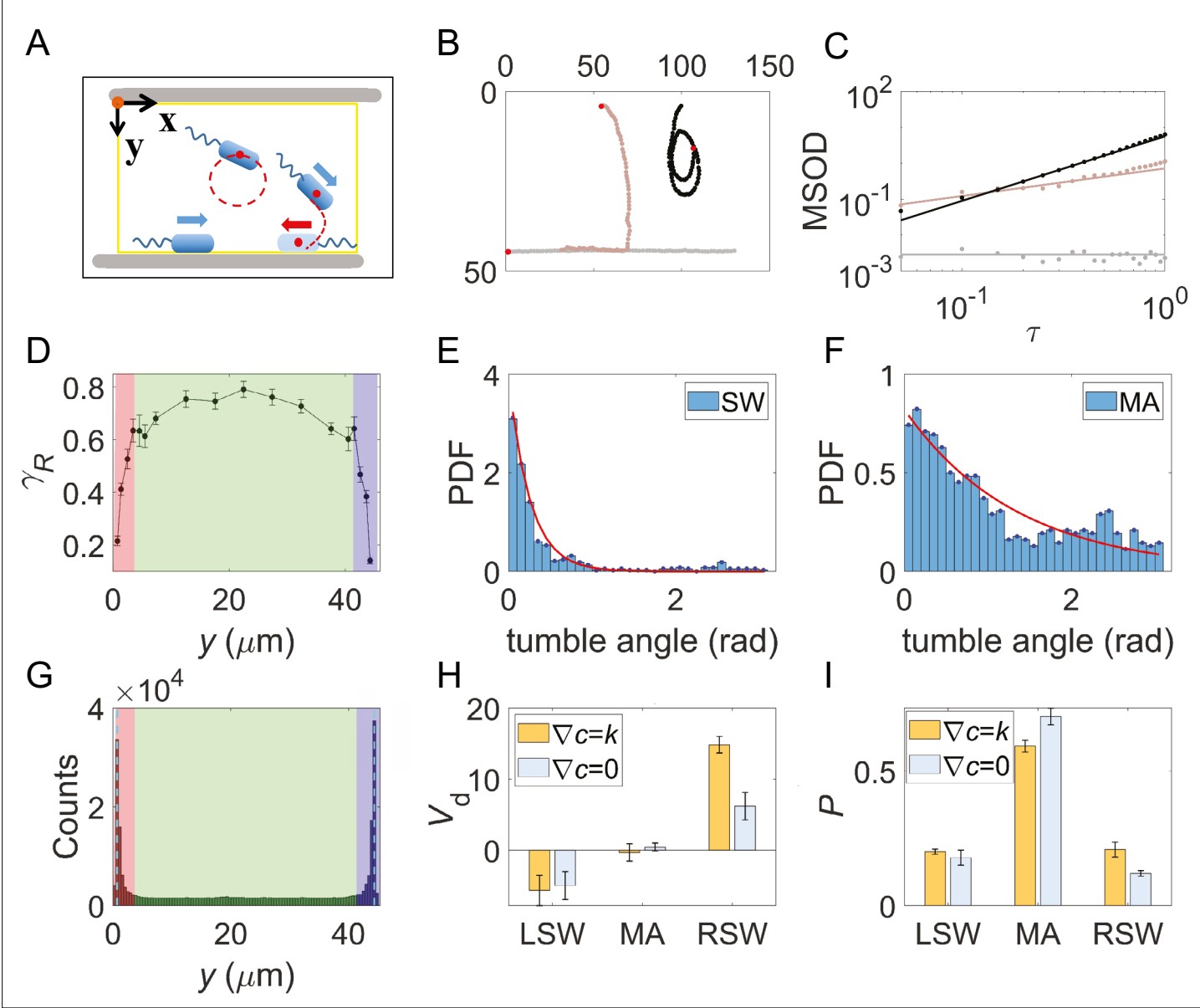

**Figure 2.** Characterization of bacterial motion states and their dependence on wall proximity. (**A**) Schematic drawing of three typical motion states. (**B**) Trajectories for three typical motion states. Red dots represent the start of each trajectory. (**C**) Calculating the rotational exponent for each trajectory, $\gamma_R$, by fitting the mean-squared orientational displacement $\langle MSOD \rangle$. Dots were experimental data calculated from the tracks in (**B**). Solid lines were fitting results with $\langle MSOD \rangle = C_R \tau^{\gamma_R}$. The fitted $\gamma_R$ values are 0, 0.77, and 1.80 for gray, brown, and black tracks, respectively. (**D**) Relationship between the rotational exponent $\gamma_R$ and the mean $y$ position of bacterial trajectories. Each data point represents the mean of at least 50 trajectories. Errors denote standard error of the mean (SEM). Red and purple shaded areas represent the region 3 μm from the left sidewall (LSW) and right sidewall (RSW), respectively. Green shaded area represents the middle area (MA) region. (**E and F**) Normalized distribution of tumble angle for tracks along the sidewalls (**E**) and in the MA region (**F**), respectively. Red solid lines are fitting results with an exponential function: $y = a * \exp(-x/b)$, where $a = \frac{1/b}{1-e^{-\pi/b}}$ is a normalization constant. (**G**) Distribution of bacteria along the $y$-axis in the lanes with a width of 44 μm. The shades of different colors denote the same meaning as in (**D**). (**H**) Drift velocity of bacterial cells in the three regions. (**I**) Proportions of bacterial trajectories in the three regions. The datasets used in panels (**H**) and (**I**) are from the assays in *Figure 1C*. Errors are standard deviations (SDs) of 6 movies.

The online version of this article includes the following figure supplement(s) for figure 2:

**Figure supplement 1.** Typical examples of collision trajectories between bacteria and sidewalls.

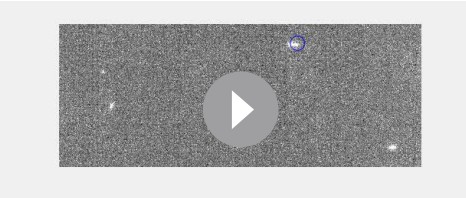

**Video 2.** An example of a large reorientation angle during tumbling in the left sidewall (LSW) of a 44-μm-wide lane under a linear gradient of L-aspartate. The source channel is on the right side. The cell of interest is marked with a blue circle.

https://elifesciences.org/articles/102686/figures#video2

## The dependence of chemotactic performance on lane width

Having established that *E. coli* chemotaxis on lane surfaces is closely related to sidewall confinement, we investigated chemotaxis in lanes of different widths to mimic bacterial chemotactic behavior in interstitial tissues or tubes of varying sizes. We compared six lanes with widths of 6 μm, 8 μm, 10 μm, 15 μm, 25 μm, and 44 μm. Drift velocity was calculated using the same method as in *Figure 1C*, with results shown in *Figure 3A*. The mean drift velocities were 1.8±1.3 μm/s, 7.5±1.4 μm/s, 4.6±1.7 μm/s, 2.5±0.5 μm/s, 2.6±0.4 μm/s, and 1.6±0.3 μm/s for lanes of width 6 μm, 8 μm, 10 μm, 15 μm, 25 μm, and 44 μm, respectively. Bacteria exhibited optimal chemotaxis (highest drift velocity) in lanes that were 8 μm wide. To rule out any potential ratchet effect caused by the asymmetric entrances, we performed control measurements using a setup with straight entrances; these results confirmed our findings and are shown in *Figure 3—figure supplement 1*.

Noting that this optimal width is of the same order of magnitude as the radius of bacterial circular swimming on a solid surface, we measured the radius of circular swimming for cells in the MA region. As shown in *Figure 3B*, the peak radius in the distribution of circular swimming radii is ~10 μm, which is close to the lane width supporting optimal chemotaxis.

It has been reported that bacteria swimming in capillaries of different radii possess different swimming speeds (*Ping et al., 2015*). Therefore, we calculated the cell swimming speeds for all lane widths. As shown in *Figure 3C*, swimming speed remains relatively constant across different lane widths. This indicates that the optimized chemotaxis is not a result of varying swimming speeds.

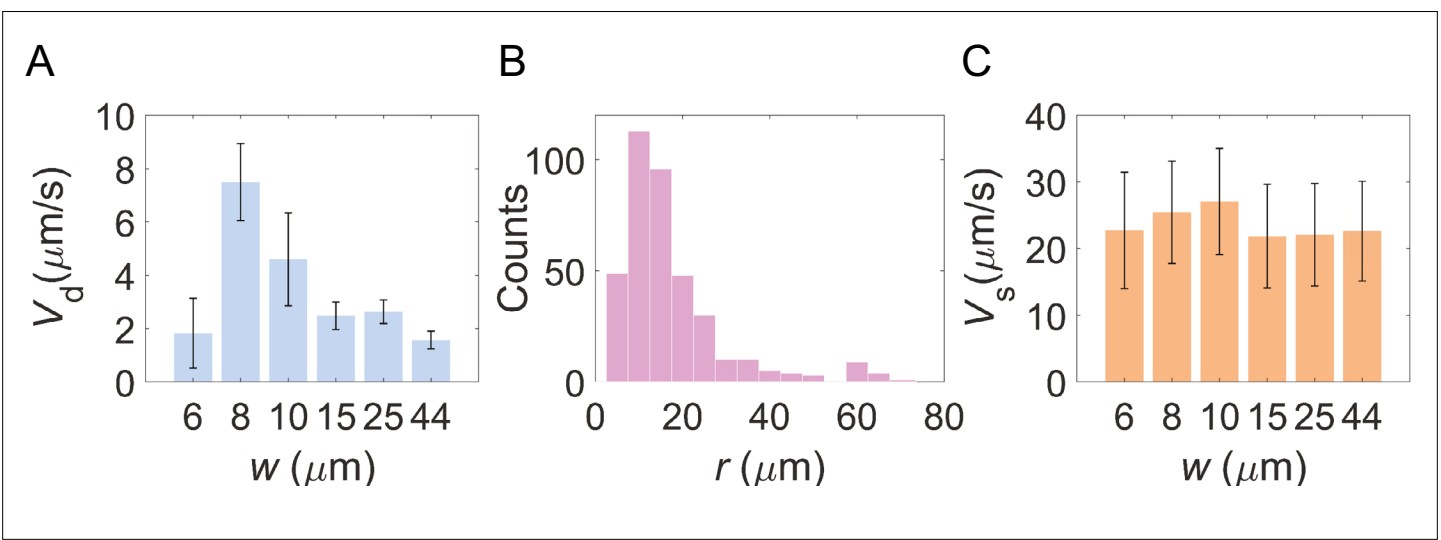

**Figure 3.** Chemotactic performance of bacteria in lanes of different widths. (**A**) Drift velocity of cells in lanes with different widths. The drift velocities and standard deviations for lanes with widths of 6 μm, 8 μm, 10 μm, 15 μm, 25 μm, and 44 μm are 1.8±1.3 μm/s, 7.5±1.4 μm/s, 4.6±1.7 μm/s, 2.5±0.5 μm/s, 2.6±0.4 μm/s, and 1.6±0.3 μm/s, respectively. The corresponding numbers of trajectories are 318, 302, 254, 1634, 2043 and 3206, respectively. (**B**) The distribution of the radius of circular swimming from 382 trajectories in the middle area (MA) regions. The peak value of the radius is ~10 μm. (**C**) The swimming speed of cells in lanes with different widths. The mean swimming speeds and standard deviations for lanes with widths of 6 μm, 8 μm, 10 μm, 15 μm, 25 μm, and 44 μm are 22.7±8.7 μm/s, 25.4±7.7 μm/s, 27.1±7.9 μm/s, 21.9±7.8 μm/s, 22.1±7.7 μm/s, and 22.6±7.5 μm/s, respectively. The number of trajectories per lane width is the same as **A**.

The online version of this article includes the following figure supplement(s) for figure 3:

**Figure supplement 1.** The relationship between chemotactic drift velocity and channel width measured using a control setup with straight entrances.

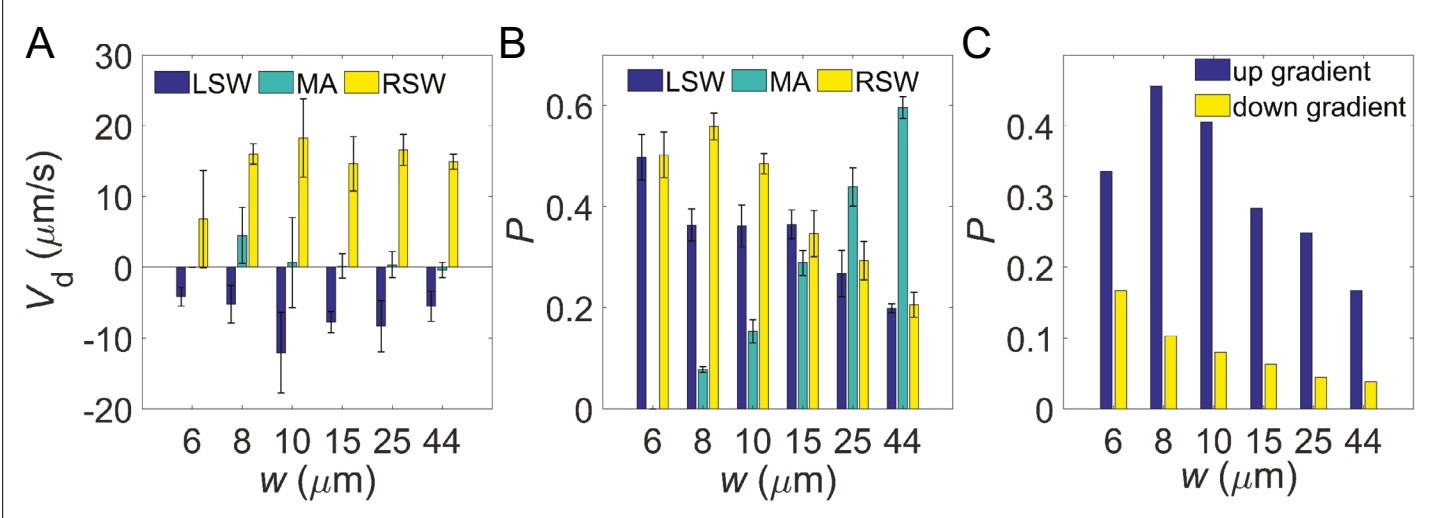

**Figure 4.** Bacterial behavior in distinct regions of lanes with different widths. (**A**) Mean drift velocities in the left sidewall (LSW), middle area (MA), and right sidewall (RSW) regions of lanes with different widths. The errors are standard deviations (SDs) of different movies. (**B**) The proportion of bacterial cells in the LSW, MA, and RSW regions of lanes with different widths. Errors represent SD of different movies. There are 3 movies for 6 μm, 8 μm and 10 μm-wide lanes, and 6 movies for 15 μm, 25 μm and 44 μm-wide lanes. (**C**) The proportion of right sidewall swimming up-gradient (RSW-UG) (cells swimming up-gradient in the RSW) and right sidewall swimming down-gradient (RSW-DG) (cells swimming down-gradient in the RSW) for lanes with different widths.

The online version of this article includes the following figure supplement(s) for figure 4:

**Figure supplement 1.** The relationship between cell enrichment and lane width (*w*).

### The optimal chemotaxis resulted from more cells moving toward the attractant along the RSW

To identify the underlying mechanism of optimal lane width for chemotaxis, we evaluated the drift velocity for cells in the LSW, MA, and RSW regions separately. The results are shown in *Figure 4A*. Cells in the MA region did not exhibit effective drift, while LSW and RSW cells demonstrated negative and positive drift, respectively, consistent with the findings in *Figure 2H*. The drift velocity for RSW cells increased with lane width and remained relatively constant above a lane width of 8 μm, thus not exhibiting a highest value in the 8-μm-wide lanes. Therefore, the width-dependent optimal chemotaxis is not attributable to an optimal drift velocity for RSW cells.

We then sought a comparison of cell proportions in the three areas, as plotted in *Figure 4B*. The proportion of cells in the MA region increased gradually with lane width due to the increase in area. Notably, the largest proportion of RSW cells appears in the 8-μm-wide lane. In other words, the lane width-dependent optimal chemotaxis primarily resulted from changes in cell numbers on the sidewalls. The relation between cell enrichment and lanes' width was plotted in *Figure 4—figure supplement 1*.

To further investigate changes in up- and down-gradient cell numbers in the RSW region, we calculated the proportion of RSW-UG (cells swimming up-gradient in the RSW) and RSW-DG (cells swimming down-gradient in the RSW) cells. As shown in *Figure 4C*, more cells moved toward the attractant in the 8-μm-wide lane. Thus, the optimal chemotaxis in the 8-μm-wide lane resulted from a higher number of cells moving toward the attractant along the RSW.

### Simulation of *E. coli* chemotaxis within lane confinements

To explore the relationship between the circular swimming of bacteria on lane surfaces and optimal chemotaxis, we simulated two-dimensional chemotaxis of bacteria with varying circular swimming radii on lanes of different widths in a steady linear gradient of L-aspartate.

In our simulation, cells were treated as self-propelled particles performing a random walk in a run-and-tumble mode within a two-dimensional space. Cells could be in either a run or tumble state, determined by the intracellular chemotaxis signal. During a run, cells swam smoothly at a constant

speed of $v_0 = 20$ µm/s. The effect of the bottom surface on bacterial swimming was modeled by adding a constant angular velocity term to the swimming direction:

$$\theta\left(t+\Delta t\right) = \theta\left(t\right) + \left(-\frac{v_0}{r}\right)\Delta t + \sqrt{2D_r\Delta t}\, n\left(0, 1\right)$$

where $\theta$ denotes the angle between the swimming direction and the positive x-axis, $\Delta t$ is the time step, $r$ is the radius of circular swim, $D_r$ is the rotational diffusion coefficient, and $n(0, 1)$ represents a random number drawn from a standard normal distribution. When cells collided with sidewalls, their velocities instantly aligned with the sidewall (*Tao et al., 2025*) and could potentially leave the sidewall after their next tumble. During a tumble, cells stopped moving, and their swimming direction changed. The change in $\theta$ was selected from the distribution of tumble angles observed in *Figure 2E and F* for different areas.

Initially, 100 cells were evenly distributed on the surface of a lane measuring 160 µm in length and $w$ in width. A linear concentration gradient of L-aspartate, identical to the experimental conditions, was established along the x-axis according to:

$$c\left(x\right) = 0.05x + 22,$$

with the concentration $c$ in µM and the position $x$ in µm. The simulation for each set of parameters ran for 150 s with a time step of 0.05 s and was repeated 50 times.

*Figure 5A* illustrates the relationship between chemotactic drift velocity and lane width for cells with a 10-µm-radius circular swim. The highest drift velocity was observed at a lane width of 8 µm. *Figure 5B* shows the proportions of cells in the three regions (LSW, MA, RSW) for different lane widths, presenting profiles similar to those observed experimentally in *Figure 4B*. The drift velocity for three regions was shown in *Figure 5—figure supplement 1*. Results for different swimming radii are presented in *Figure 5C*, demonstrating that the optimal lane width for chemotaxis is closely related to the radius of circular swimming. It is important to note that our model treats bacteria as active particles and does not explicitly account for the steric exclusion of the flagella and cell body. In reality, these effects would restrict the cells' ability to turn freely in narrow lanes (≤6 µm), thereby reducing the cell density mismatch between the LSW and RSW and decreasing the drift velocity compared to the simulation. Consequently, the distinct behaviors of the LSW and RSW populations observed in the simulation at very small widths (*Figure 5B*) reflect the idealized nature of our model, specifically the absence of steric hindrance.

As shown in *Figure 5D*, the relationship between optimal lane width ($w$) and circular swimming radius ($r$) was fitted with a linear function $w = kr$, with $k = 0.66 \pm 0.03$. This indicates a clear proportional relationship between the lane width for optimal chemotaxis and the circular swimming radius of bacteria.

## Geometrical analysis of optimal lane width for chemotaxis

We have established that the optimal lane width for chemotaxis (8 µm) primarily results from a higher proportion of cells in the RSW region swimming up-gradient. Additionally, this optimal width is directly proportional to the circular swimming radius of the bacteria. To further elucidate the underlying mechanism of this phenomenon, we employed geometrical analysis.

We assumed an initial uniform distribution of cells across the lane width. As shown in *Figure 6A–C*, we divide the width of the lane into three scenarios:

1. $0 < w \le r$: For a cell at any point P in the channel, its swimming direction can be arbitrary. However, only cells swimming in directions between $\theta_1$ and $\theta_2$ (highlighted in green) can swim up-gradient in the RSW after their first collision with the sidewall. The proportion of these cells can be calculated as

$$\frac{1}{w}\int_0^w \frac{\theta_1+\theta_2}{2\pi}\,dy \quad = \frac{1}{2\pi w}\int_0^w dy\left(\arccos\frac{r-w+y}{r} + \arccos\frac{y}{r}\right)$$
$$= \frac{1}{2\pi m}\left[-\left(1-m\right)\arccos\left(1-m\right) + \sqrt{1-\left(1-m\right)^2} + m\arccos m - \sqrt{1-m^2} + 1\right],$$

where $m = w/r \le 1$.

2. $r < w \le 2r$: Only the cells in the region $y < r$ and with velocity direction between $\theta_1$ and $\theta_2$ can swim up-gradient in the RSW after their first collision with the sidewall. The proportion of cells in this region is:

$$\frac{1}{w}\int_0^r \frac{\theta_1+\theta_2}{2\pi}\,dy = \frac{1}{2\pi w}\int_0^r dy\left(2\arccos\frac{y}{r}\right) = \frac{1}{\pi m}.$$

3. $w > 2r$: The proportion of cells that can swim up-gradient in the RSW is calculated the same as in case 2:

$$\frac{1}{w}\int_0^r \frac{\theta_1+\theta_2}{2\pi}\,dy = \frac{1}{\pi m}.$$

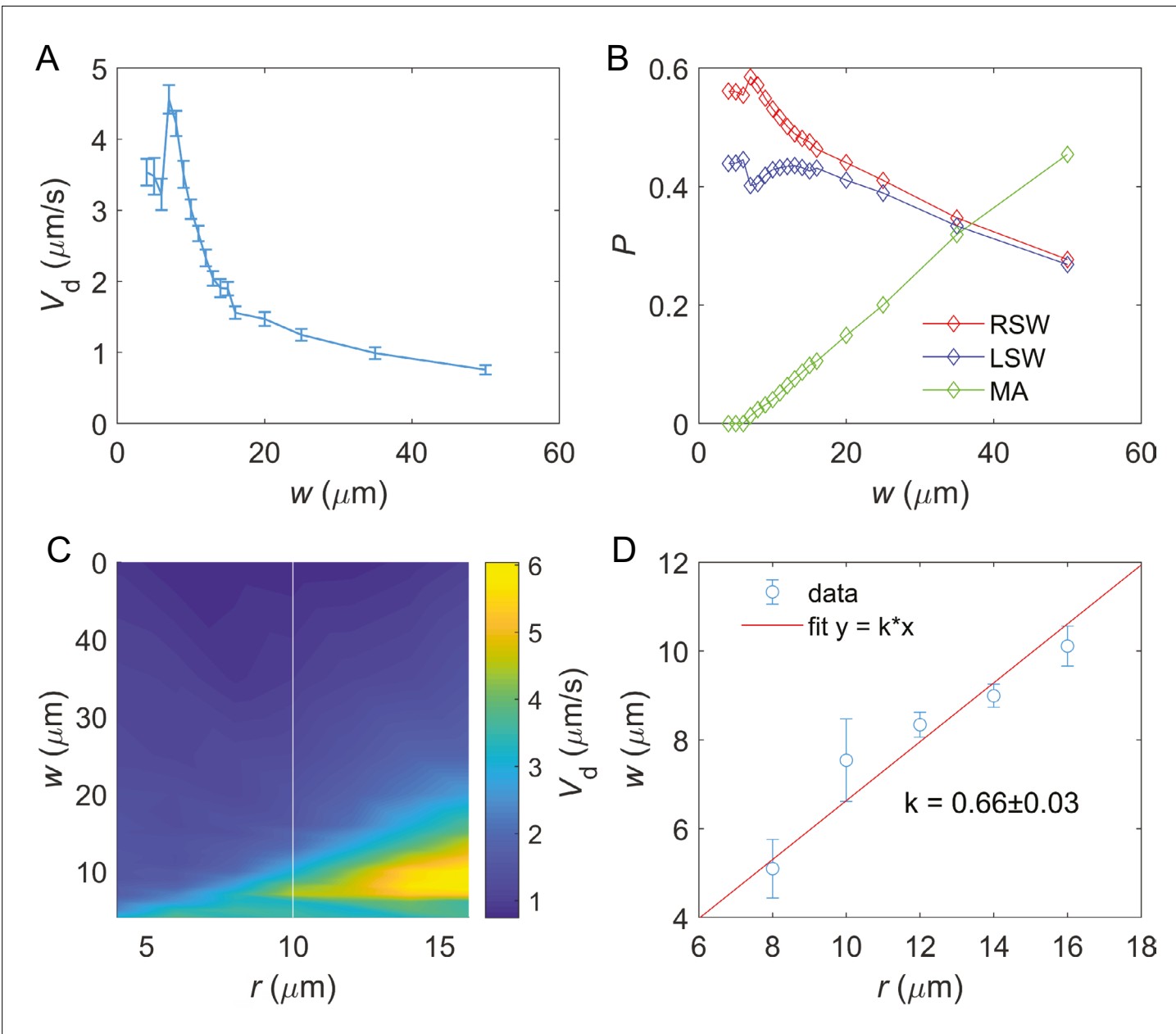

**Figure 5.** Simulation of bacterial chemotaxis on the surface of lanes. (**A**) The relationship between drift velocity $V_d$ of cells with 10-μm-radius circular swim and the lane width $w$. Errors denote standard error of the mean (SEM) calculated from 50 simulations. (**B**) The proportion of bacterial trajectories in the left sidewall (LSW), middle area (MA), and right sidewall (RSW) regions of lanes with different widths, for cells with 10-μm-radius circular swim. (**C**) The drift velocity $V_d$ of cells with different radii of circular swim in lanes of different widths. (**D**) The relationship between circular swim radius and lane width for optimal chemotaxis (maximal drift velocity) in (**C**). The widths were extracted by fitting the peak value of $w$ in **A** with a Gaussian function. The red solid line represents a linear fit. The slope is 0.66±0.03. Errors denote standard deviation (SD).

The online version of this article includes the following figure supplement(s) for figure 5:

**Figure supplement 1.** Mean drift velocities in the left sidewall (LSW), middle area (MA), and right sidewall (RSW) regions of lanes with different widths from simulations.

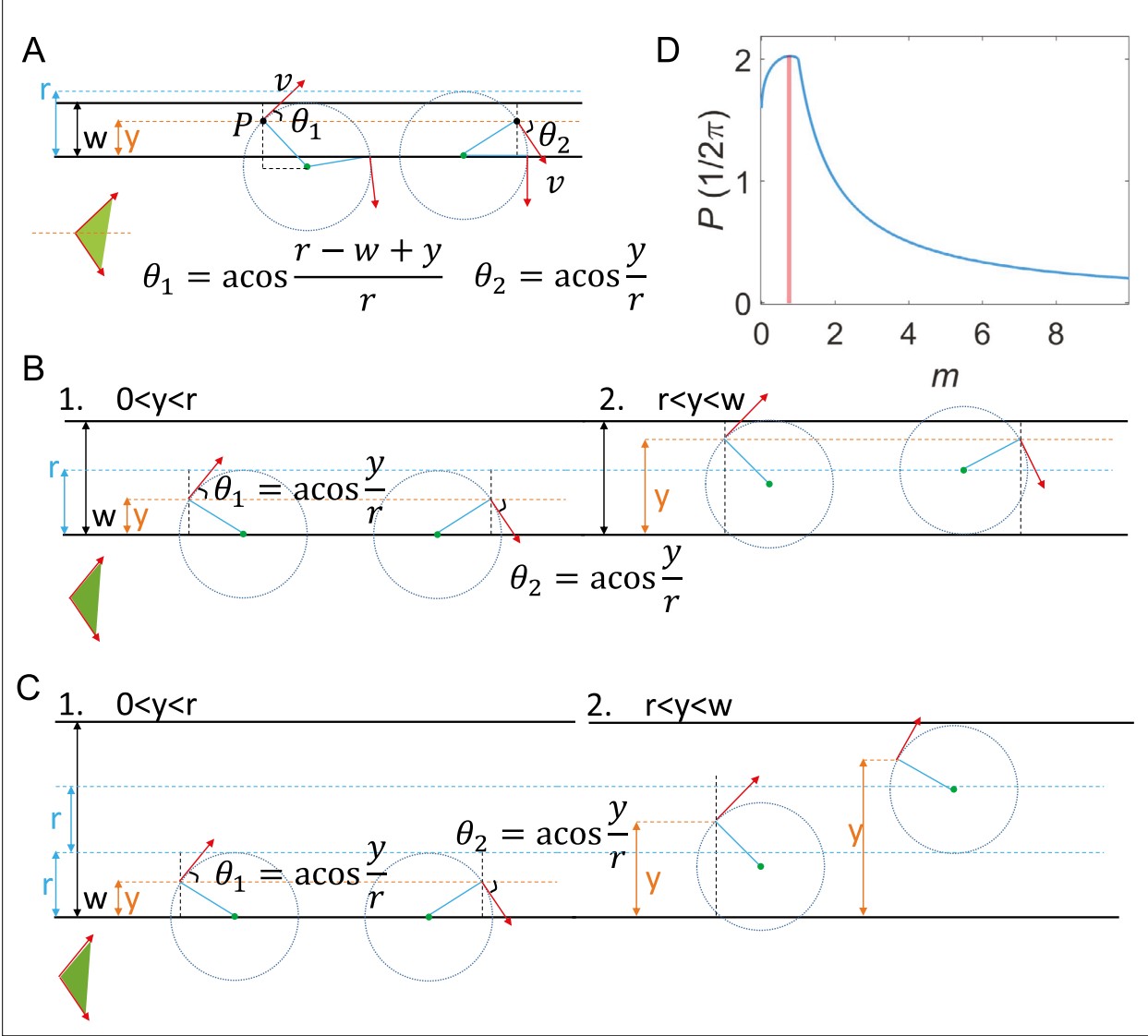

**Figure 6.** Geometrical analysis of optimal lane width for chemotaxis. (**A**) Case 1: $0<w \leq r$. (**B**) Case 2: $r<w \leq 2r$. (**C**) Case 3: $w>2r$. The red arrows denote the velocity direction. The black solid lines represent the sidewalls. Dashed circles are trajectories. The up-gradient direction is along the positive x-axis. The green shade areas label the swim direction of cells that can swim up-gradient in the right sidewall (RSW). (**D**) Relationship between probability ($P$) of cells swimming up-gradient in the RSW and $m$, where $m=w/r$. Red shaded area denotes $m$ for maximal $P$, $m \in (0.7, 0.8)$.

The complete relationship between the proportion of cells that can swim up-gradient in the RSW ($P$) and $m$ is plotted in **Figure 6D**. The maximum $P$ occurs when $m \in (0.7, 0.8)$, marked by the red shaded area. This value is consistent with the slope $k$ in **Figure 5D** and results in the optimal width of about 8 µm in **Figure 3A** for a circular swimming radius of 10 µm.

Crucially, the observed asymmetry arises because the angular ranges that geometrically facilitate RSW-UG accumulation (**Figure 6A–C**) coincide with the up-gradient direction. Because cells moving up-gradient experience suppressed tumbling, they can maintain the steady circular trajectories required to reach and align with the RSW. Conversely, while a pure geometric analysis suggests a symmetric potential for LSW-DG accumulation, these trajectories coincide with the down-gradient direction. Consequently, these cells experience enhanced tumbling, which distorts the circular trajectories. This prevents them from effectively reaching the LSW and also increases the probability of them leaving the wall.

# Discussion

Bacterial behavior in confined environments differs significantly from that in free environments. Previous studies have reported that cells tend to swim in circles and accumulate near a solid-liquid interface, leading to diminished chemotaxis. However, recent research on chemotaxis of swarm cells suggests that crowded environments may enhance bacterial chemotaxis on surfaces. In this study, we examined the chemotaxis of *E. coli* on the bottom surface of microfluidic lanes, which simulate the spatial constraints imposed by neighboring cells in a swarm, exploring the effects of lane confinement on bacterial chemotaxis.

We found that in the presence of sidewalls, cells that would otherwise have been unable to exhibit chemotaxis showed significant chemotactic behavior in a linear gradient of L-aspartate. Cells in different regions of the lanes demonstrated distinct motion behaviors. In the MA region, cells showed no drift, similar to observations in chemotaxis near surfaces. Conversely, cells exhibited positive and negative drift when in the RSW and LSW regions, respectively.

Further investigation into chemotaxis in lanes of varying widths demonstrated that cells exhibited the highest chemotactic drift velocity in lanes that were 8 μm wide. This width is close to the radius of bacterial swimming circles. By analyzing the proportion and drift velocity of cells in different regions, we determined that this optimal lane width was primarily due to the higher proportion of cells swimming up-gradient along the RSW.

To explore the relationship between chemotactic drift velocity and circular swimming radius of bacteria, we conducted systematic simulations of *E. coli* chemotaxis with different circular swimming radii in lanes of varying widths under a linear concentration gradient of L-aspartate. These simulations explicitly incorporated the chiral clockwise circular motion of surface-swimming bacteria and the alignment effect induced by sidewall collisions. We found that optimal chemotaxis occurred when the swimming radius was close to the lane width, with $w=0.66r$. This suggests a significant proportional relationship between the optimal lane width for chemotaxis and the circular swimming radius of bacteria.

To elucidate the mechanism underlying this phenomenon, we employed geometric analysis. We demonstrated that bacteria in lanes could swim up-gradient in the RSW with maximal probability when the lane width was 0.7–0.8 times the radius of circular swimming. This finding is consistent with the experimental results presented in *Figure 4C*.

Our findings have important implications for the screening of single-celled swimming organisms and the study of restricted chemotactic behavior. This research provides insights into how confinement affects bacterial chemotaxis and may inform the design of microfluidic devices for bacterial manipulation and analysis.

Bacteria frequently inhabit confined spaces such as soil pores, sediment interstices, rock crevices, biological tissues, and hydrogels. In these constrained environments, bacterial behavior often deviates significantly from that observed in unconfined conditions, typically resulting in reduced chemotactic ability. However, our findings challenge this notion, suggesting that certain types of environmental confinement may actually enhance bacterial chemotaxis. This discovery could contribute to our understanding of microbial ecology across various habitats, including soil, oceans, and the human body.

Furthermore, these insights could have potential applications. In biotechnology, optimizing confinement conditions could improve the efficiency of bioremediation processes or the production of microbial products. In medicine, understanding how pathogens navigate confined spaces within the body could lead to novel strategies for preventing or treating infections.

# Materials and methods

**Key resources table**

| Reagent type (species) or resource | Designation | Source or reference | Identifiers | Additional information |
|---|---|---|---|---|
| Strain, strain background (*Escherichia coli*) | Wild-type AW405 | Howard Berg Lab (*Armstrong et al., 1967*) | HCB1 | Also known as AW405 |
| Recombinant DNA reagent | pTrc99a-mCherry (plasmid) | Junhua Yuan Lab (*Tian et al., 2021*) | pMT1 | mCherry |
| Chemical compound, drug | IPTG | Sigma-Aldrich | CAT#I6758 | |

*Continued on next page*

*Continued*

| Reagent type (species) or resource | Designation | Source or reference | Identifiers | Additional information |
|---|---|---|---|---|
| Chemical compound, drug | L-Aspartic acid | BBI | CAT#A600091 | |
| Chemical compound, drug | Fluorescein | Sigma-Aldrich | CAT#46955 | |
| Chemical compound, drug | Agarose | Sigma-Aldrich | CAT#V900510 | |
| Software, algorithm | Matlab R2018b | MathWorks | RRID:SCR_001622 | |
| Software, algorithm | Fiji | Fiji | RRID:SCR_002285 | |
| Software, algorithm | Custom script | *Yue and Zhang, 2026* | https://github.com/cjyue2024/eLife_enhanced-chemotaxis | |

## Strain and plasmids

The plasmid pTrc99a-mCherry expresses mCherry under an IPTG-inducible promoter. The wild-type *E. coli* K12 strain AW405 (HCB1) (*Armstrong et al., 1967*) transformed with pTrc99a-mCherry (*Tian et al., 2021*) was used for fluorescent tracking of cell trajectories. Cells were streaked on a Petri dish containing 1.5% agar and lysogeny broth (1% tryptone, 0.5% NaCl, 0.5% yeast extract, supplemented with the appropriate antibiotic: 100 µg/ml ampicillin). A single colony was inoculated into lysogeny broth and grown overnight at 33°C with 200 rpm rotation. The saturated culture was then diluted 1:100 (100 µl in 10 ml) into tryptone broth (1% tryptone, 0.5% NaCl, supplemented with the appropriate inducer and antibiotic: 0.1 mM IPTG, 100 µg/ml ampicillin) and grown at 33°C with 200 rpm rotation to $OD_{600}$=0.53. Cells were then harvested from culture media by centrifugation at $1.2 \times g$ for 6 min at room temperature. The pellet was resuspended by gently mixing in motility medium (10 mM potassium phosphate, 0.1 mM ethylenediaminetetraacetic acid, 10 mM lactic acid, and 1 µM methionine at pH 7.0). The cells were washed three times to replace growth medium with motility medium. One hour after fluorescence maturation in a 33°C incubator, cells were placed in a 4°C refrigerator ready for microfluidic experiments.

## Microfluidic device design and fabrication

A hydrogel-based microfluidic device containing lanes of different width was designed using L-edit based on previous work (*Ping et al., 2015*). The silicon mold with the positive relief features was fabricated using the standard soft lithography technique performed in our laboratory. First, a 4-inch diameter silicon wafer was spin-washed sequentially using water, acetone, methanol, and isopropanol. The wafer was thoroughly dried on a 150°C hot plate to ensure that organics had volatilized completely. Then, the wafer was treated with air plasma (Harrick plasma, PDC-002, 30W) for 10 min before the SU-8 2025 photoresist (Microchem) was spin-coated according to the products protocols. The spin-coated wafer was placed on a hot plate at 65°C for 3 min followed by 6 min at 95°C, then exposed to UV light using a maskless lithography machine (Heidelberg, µMLA) at 800 mJ/cm$^2$ with defocus set to 0. The post-exposure bake was performed at 65°C for 2 min followed by 6 min at 95°C. Development procedure lasted about 6 min with constant shaking in the SU-8 developer. The wafer was rinsed with fresh developer followed by isopropanol. Finally, an approximately 30-µm-high photoresist layer was created as the master mold. The mold was fixed on a glass Petri dish using double-sided tape.

PDMS and curing agent were mixed with a 10:1 ratio (Sylgard 184, Dow Corning) and centrifuged at 3500 rpm for 5 min to remove air bubbles. The mixture was poured over the mold in a glass Petri dish and then placed at 4°C for an hour to avoid interference from very small air bubbles. The PDMS was cured at 80°C for 3 hr, then separated from the wafer and cut into pieces, and holes were punched for inlets and outlets. The punched PDMS was cleaned with adhesive tape. The cleaned PDMS was bonded to a glass slide after air plasma treatment (Harrick plasma, PDC-002, 30W). The fabricated device was baked at 80°C overnight to establish covalent bonding.

The hydrogel agarose was used at 2% (wt/vol) concentration in motility medium to create diffusion-permeable walls. Agarose was slowly injected through the inlet port using a syringe pump at a constant flow rate of 2 µl/min, and the injection process was carried out in a drying oven at a constant temperature of 68°C.

## Microscopy and data acquisition

The PDMS microfluidic chip was placed on a Nikon Ti-E inverted optical microscope equipped with a specific filter set, a 20× objective (Nikon, CFI S Plan Fluor ELWD, 20×, NA 0.45), and a CMOS camera (Hamamatsu, ORCA-Flash4.0, pixel size = 6.5 μm). We set the objective lens's correction collar to 1.2 mm to match the glass slide thickness. A halogen lamp serves as the bright-field light source to help locate the observation area, while a xenon lamp serves as the excitation light source for fluorescence experiments. The depth of field is 3.7 μm.

## Gradient profile calibration

Fluorescein solution (Sigma) was used for visualizing the chemical concentration field in the microfluidic device. 100 μM fluorescein and motility medium were flowed into the source and sink channels, respectively, at a constant flow rate of 10 μl/min. One fluorescent image was captured per minute. To avoid the impact of photobleaching on the results, we only illuminated the sample during the imaging process.

## Chemotaxis assays

The L-aspartic acid and motility medium solutions were respectively introduced into the source and sink channels and left for more than 1 hr to allow L-aspartic acid to diffuse through the agarose wall and form a stable concentration gradient field. Then, bacteria were introduced into the cell channel at a rate of 5 μl/min using a syringe pump for a duration of 90 s. After the bacteria injection was complete, the microvalves were closed to seal the cell channel. Bacteria then performed chemotaxis due to the perception of the concentration gradient field. After waiting for 2 min, the shutter was opened to illuminate the fluorescent bacteria with the excitation light, and video recording began. During the video acquisition process, fluorescence excitation light was continuously illuminated onto the bacteria. For each recording, 9600 frames (2048×2048 pixels) were captured at a frame rate of 20 fps and an exposure of 50 ms. Although the long-term illumination caused a decrease in the brightness of the fluorescent bacteria, the position information of the bacteria could still be accurately distinguished within the video duration (8 min).

## Data analysis

### Tracking

Image analysis was performed using TrackMate plugin within Fiji. We chose the Log detector with an estimated blob diameter of 70 μm and a threshold of 800 to effectively recognize bacteria, based on the diameter of bacteria and the use of a 20× objective. Then, we used the LAP Tracker and set the maximum distance for frame-to-frame linking to 50 μm.

### Swimming speed

The velocity is calculated using the fourth-order central difference method by *Turner et al., 2016*:

$$\vec{v}_i = \frac{8\left(\vec{x}_{i+1} - \vec{x}_{i-1}\right) - \left(\vec{x}_{i+2} - \vec{x}_{i-2}\right)}{12T}.$$

The magnitude of the velocity is given by $v_i = \left|\vec{v}_i\right|$. The speed of each trajectory is defined as the average of $v_i$. The swimming speeds of bacteria in lanes of different widths are represented by the mean and standard deviation of all trajectories.

### Calculation of the radius of circular motion on the solid surface

A total of 382 trajectories with a rotational exponent greater than 1.6 in the MA region of the $w$=44 μm lanes were selected for analysis. The trajectories were smoothed using the Savitzky-Golay filter with a window size of 5. The least squares method was then applied to solve the linear equations, yielding the center and radius of the fitted circles. The principle of this process is briefly described as follows.

The standard equation of a circle can be written as:

$$(x-a)^2 + (y-b)^2 = r^2,$$

where $(a, b)$ are the coordinates of the circle's center, and $r$ is the radius. Expanding and rearranging the standard equation of a circle, it can be rewritten as:

$$x^2 + y^2 = 2ax + 2by + c,$$

where $c = r^2 - a^2 - b^2$.

Rewriting the above equation in matrix form, we have:

$$B = A \cdot Coefficients$$

where

$$B = x^2 + y^2 = \begin{bmatrix} x_1^2 + y_1^2 \\ x_2^2 + y_2^2 \\ \vdots \\ x_n^2 + y_n^2 \end{bmatrix}$$

is the response vector, and

$$A = \begin{bmatrix} x & y & 1 \end{bmatrix} = \begin{bmatrix} x_1 & y_1 & 1 \\ x_2 & y_2 & 1 \\ \vdots & \vdots & \vdots \\ x_n & y_n & 1 \end{bmatrix}$$

is the design matrix. The coefficients are given by:

$$Coefficients = \begin{bmatrix} 2a \\ 2b \\ c \end{bmatrix} = \left(A^T A\right)^{-1} \left(A^T B\right).$$

## Tumble angle

A tumble is defined as points where the velocity is lower than the average velocity of the trajectory $\langle v_i \rangle$ and shows a substantial drop from the previous time point. Specifically, a tumble is identified when:

$$\{v_{t1}, \cdots, v_{tn}\} < \langle v_i \rangle,$$

$$\{v_{t0} - v_{t1}, \ldots, v_{tn-1} - v_{tn}\} \langle v_i \rangle \cdot \alpha,$$

where $n \geq 2$, $\alpha = 1/6$ is the threshold, and $t_1$ and $t_n$ represent the start and end times of a tumble, respectively.

## Simulation of bacterial chemotaxis within lanes

In our simulation, we introduced the effects of bottom surface and sidewalls into the two-dimensional chemotaxis of *E. coli*. The simulations were carried out in a rectangular area of size 160 µm×$w$, where $w$ denotes the lane width. Cells were treated as self-propelled particles. They could sense and adapt to the ligand in the surrounding environment via the intracellular chemotaxis signaling pathway, which could be described by a coarse-grained model of bacterial chemotaxis (*Jiang et al., 2010*; *Tu et al., 2008*):

$$a = \frac{1}{1 + \exp\left(N\left(\alpha(m - m_0) + \ln \frac{1 + \frac{c}{K_{off}}}{1 + \frac{c}{K_{on}}}\right)\right)},$$

$$\frac{dm}{dt} = k_R \left(1-a\right) - k_B a,$$

where $a$, $m$, and $c$ represent the activity of the receptor-kinase complex, the methylation level of receptors, and the ligand concentration, respectively. $N$ denotes the number of chemoreceptors in a Monod-Wyman-Changeux cluster (*Vladimirov et al., 2008*). The values of parameters we used were the same as before (*Ren et al., 2023*; *Tian et al., 2021*; *Vladimirov et al., 2008*): $N=4.6$, $K_{off} = 1.7$ µM, $K_{on} = 12$ µM, $\alpha=1.7$, $m_0=1.0$, $k_R = 0.005$ s$^{-1}$, $k_B = 0.010$ s$^{-1}$.

The concentration of CheY-P, which changes due to the receptor-kinase phosphorylation on CheY, could be calculated by $Yp = 7.86a$. Consequently, the CW bias ($B$) of flagellar motors would also vary (*Cluzel et al., 2000*):

$$B = \frac{\left(Yp\right)^{10.3}}{\left(Yp\right)^{10.3} + 3.1^{10.3}}.$$

It has been reported that the average run time in the near-surface region for wild-type cells is ~2.0 s (corresponding to a run-to-tumble rate of ~0.5 s$^{-1}$) (*Molaei et al., 2014*), and the steady-state motor CW bias is ~0.15 without a gradient. The run-to-tumble rate $k$ was set to be equivalent to the motor CCW-to-CW switching rate. As this rate increases linearly with motor bias $B$ (*He et al., 2016*), it was calculated using $k = \frac{B}{0.31}$ s$^{-1}$. The switching rate from tumble to run was 5 s$^{-1}$ due to the constant tumble duration of 0.2 s (*Berg and Brown, 1972*).

In this way, the motion states of each cell at any moment are determined. During a run, cells swim smoothly with a constant speed of 20 µm/s, influenced by rotational diffusion and the effect of the bottom surface on bacterial swimming:

$$\theta(t + \Delta t) - \theta(t) = \left(-\frac{v_0}{r}\right) \Delta t + n(0, 2D_r \Delta t),$$
$$x(t + \Delta t) - x(t) = v_0 \cos\theta \, \Delta t,$$
$$y(t + \Delta t) - y(t) = v_0 \sin\theta \, \Delta t.$$

where $\theta$ is the angle between the swimming direction and the positive $x$-axis, $v_0 = 20$ µm/s is the swimming speed, $r$ is the radius of circular swimming due to the effect of the bottom surface, and $n\left(0, 2D_r\Delta t\right)$ represents a normal distribution with zero mean and variance $2D_r\Delta t$. The rotational diffusion coefficient was 0.062 rad$^2$/s (*Vladimirov et al., 2010*). During a tumble, cells stop swimming, and a tumble angle is selected from the distribution that we measured (*Figure 2E*). We also imposed a maximum detention time limit which, when combined with the variable tumble rate, results in an average wall residence time of approximately 2 s, matching our experimental observations (*Appendix 1—figure 1*). Cells will get a random $y$-coordinate at the other side when they swim through the $x$ boundary.

## Acknowledgements

This work was supported by National Natural Science Foundation of China Grants (12090053, 12474204, and 12104436), Fundamental and Interdisciplinary Disciplines Breakthrough Plan of the Ministry of Education of China (JYB2025XDXM502), University of Science and Technology of China Research Funds of the Double First-Class Initiative (YD2030002501), a grant from the Ministry of science and technology of China (2019YFA0709303), and Guizhou Provincial Major Scientific and Technological Program XKBF (2025)010.

## Additional information

### Funding

| Funder | Grant reference number | Author |
| --- | --- | --- |
| National Natural Science Foundation of China | 12090053 | Junhua Yuan |

| Funder | Grant reference number | Author |
| --- | --- | --- |
| National Natural Science Foundation of China | 12474204 | Rongjing Zhang |
| National Natural Science Foundation of China | 12104436 | Chi Zhang |
| Ministry of Education of the People's Republic of China | JYB2025XDXM502 | Junhua Yuan |
| University of Science and Technology of China | YD2030002501 | Junhua Yuan |
| Ministry of Science and Technology of the People's Republic of China | 2019YFA0709303 | Rongjing Zhang |
| Guizhou Provincial Major Scientific and Technological Program | XKBF (2025)010 | Junhua Yuan |

The funders had no role in study design, data collection and interpretation, or the decision to submit the work for publication.

### Author contributions

Caijuan Yue, Resources, Data curation, Software, Formal analysis, Validation, Investigation, Visualization, Methodology, Writing – original draft, Writing – review and editing; Chi Zhang, Conceptualization, Resources, Data curation, Software, Formal analysis, Funding acquisition, Validation, Investigation, Visualization, Methodology, Writing – original draft, Writing – review and editing; Rongjing Zhang, Conceptualization, Resources, Formal analysis, Supervision, Funding acquisition, Validation, Investigation, Methodology, Writing – original draft, Project administration, Writing – review and editing; Junhua Yuan, Conceptualization, Resources, Formal analysis, Supervision, Funding acquisition, Validation, Investigation, Visualization, Methodology, Writing – original draft, Project administration, Writing – review and editing

### Author ORCIDs

Caijuan Yue ⓘ https://orcid.org/0000-0002-1664-9897
Chi Zhang ⓘ https://orcid.org/0000-0003-3154-2657
Rongjing Zhang ⓘ https://orcid.org/0009-0008-5519-6385
Junhua Yuan ⓘ https://orcid.org/0000-0002-6437-0655

Reviewer #1 (Public review): https://doi.org/10.7554/eLife.102686.4.sa1
Reviewer #3 (Public review): https://doi.org/10.7554/eLife.102686.4.sa2
Author response https://doi.org/10.7554/eLife.102686.4.sa3

## Additional files

### Supplementary files

MDAR checklist

### Data availability

All data generated or analysed during this study are included in the manuscript and supporting files. The code is available at https://github.com/cjyue2024/eLife_enhanced-chemotaxis (copy archived at *Yue and Zhang, 2026*).

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

## Appendix 1

### Explanation of difference in drift velocity between LSW and RSW cells

We observed obvious chemotaxis of *E. coli* on the surface of lanes, closely related to cells swimming along sidewalls. As shown in *Appendix 1—figure 1*, cells swimming along sidewalls in an attractant gradient can be divided into four states: Cells on the right sidewall swimming up-gradient (RSW-UG), cells on the right sidewall swimming down-gradient (RSW-DG), cells on the left sidewall swimming up-gradient (LSW-UG), and cells on the left sidewall swimming down-gradient (LSW-DG). These cells can escape from sidewalls with rates:

$$
\begin{aligned}
k_{RSW-UG} &= B^+ k_T \\
k_{RSW-DG} &= B^- k_T + (1 - B^-) k_R \\
k_{LSW-UG} &= B^+ k_T + (1 - B^+) k_R \\
k_{LSW-DG} &= B^- k_T
\end{aligned}
$$

where $B^+$ and $B^-$ represent the probability of tumble when cells swim up-gradient and down-gradient, respectively, and $k_T$ and $k_R$ denote the escape rates during tumble and run, respectively. Cells swimming on the sidewalls experience a force $f_s$ from the bottom surface, causing them to swim to the right. RSW-UG and LSW-DG cells can swim away from the sidewalls via tumble, while RSW-DG and LSW-UG cells can escape via tumble or run. The gradient field direction ensures that $0 \le B^+ < B^- \le 1$.

Dwell times for the four states are plotted in *Appendix 1—figure 1B*, suggesting that $\tau_{RSW-UG} > \tau_{LSW-DG} > \tau_{RSW-DG} > \tau_{LSW-UG}$. Thus, $k_R > k_T > 0$. This is physically consistent: along sidewalls, running cells escape more easily via circular swimming, whereas tumbling cells are hindered by surface-suppressed reorientation and the high probability of tumbling back toward the sidewalls.

The drift velocity for cells on RSW and LSW can be calculated as:

$$
\begin{aligned}
v_d^{RSW} &= v_0 \left( \frac{1}{k_{RSW-UG}} - \frac{1}{k_{RSW-DG}} \right) = v_0 \frac{(B^- - B^+) k_T + (1 - B^-) k_R}{B^+ B^- k_T^2 + (1 - B^-) B^+ k_T k_R} \\
v_d^{LSW} &= v_0 \left( \frac{1}{k_{LSW-UG}} - \frac{1}{k_{LSW-DG}} \right) = v_0 \frac{(B^- - B^+) k_T - (1 - B^+) k_R}{B^+ B^- k_T^2 + (1 - B^+) B^- k_T k_R}
\end{aligned}
$$

where $v_0$ represents the swimming speed. Given that $0 \le B^+ < B^- \le 1$ and $k_R > k_T > 0$, we can conclude that:

$$
\begin{aligned}
v_d^{RSW} &> 0 \\
v_d^{LSW} &< 0.
\end{aligned}
$$

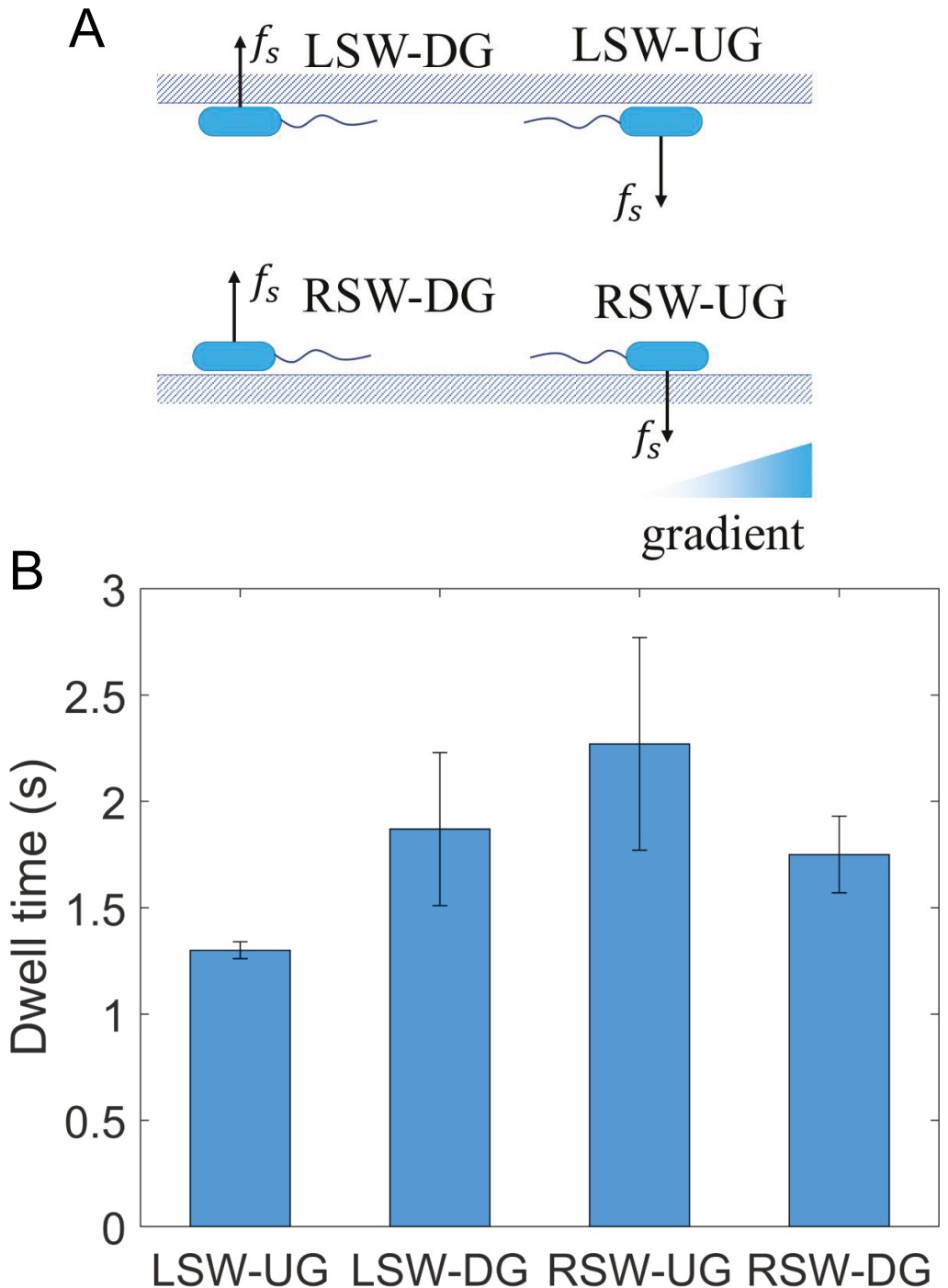

**Appendix 1—figure 1.** Motion states and dwell times of cells swimming along sidewalls. (**A**) Four motion states of cells swimming along sidewalls. Black arrows denote the direction of force generated by the bottom surface on the cell body. (**B**) Dwell times of the four motion states from experiments in 44-μm-wide lanes. The values are 1.3±0.04 s, 1.87±0.36 s, 2.27±0.5 s, and 1.75±0.18 s for LSW-UG, LSW-DG, RSW-UG, and RSW-DG, respectively. Errors represent standard deviations (SDs). The sample sizes are 173, 170, 139, and 256 for LSW-UG, LSW-DG, RSW-UG, and RSW-DG, respectively.

