## [Editor Report · eLife Assessment]

This **important** work examines the effects of side-wall confinement on chemotaxis of swimming bacteria in a shallow microfluidic channel. The authors present **convincing** experimental evidence, combined with geometric analysis and numerical simulations of simplified models, showing that chemotaxis is enhanced when the distance between the side walls is comparable to the intrinsic radius of chiral circular swimming near open surfaces. This study should be of interest to scientists specializing in bacteria-surface interactions.

---

## [Referee Report · Reviewer #1 (Public review)]

The authors show experimentally that, in 2D, bacteria swim up a chemotactic gradient much more effectively when they are in the presence of lateral walls. Systematic experiments identify an optimum for chemotaxis for a channel width of ~8µm, a value close to the average radius of the circle trajectories of the unconfined bacteria in 2D. These chiral circles impose that the bacteria swim preferentially along the right-side wall, which indeed yields chemotaxis in the presence of a chemotactic gradient. These observations are backed by numerical simulations and a geometrical analysis.

---

## [Referee Report · Reviewer #3 (Public review)]

This paper addresses, through experiment and simulation, the combined effects of bacterial circular swimming near no-slip surfaces and chemotaxis in simple linear gradients. The authors have constructed a microfluidic device in which a gradient of L-aspartate is established, to which bacteria respond while swimming while confined in channels of different widths. There is a clear effect that the chemotactic drift velocity reaches a maximum in channel widths of about 8 microns, similar in size to the circular orbits that would prevail in the absence of side walls. Numerical studies of simplified models confirm this connection.

The experimental aspects of this study are well executed. The design of the microfluidic system is clever in that it allows a kind of "multiplexing" in which all the different channel widths are available to a given sample of bacteria.

The authors have included a useful intuitive explanation of their results via a geometric model of the trajectories. In future work it would be interesting to analyze further the voluminous data on the trajectories of cells by formulating the mathematical problem in terms of a suitable Fokker-Planck equation for the probability distribution of swimming directions. In particular, this might help understand how incipient circular trajectories are interrupted by collisions with the walls and how this relates to enhanced chemotaxis.

The authors argue that these findings may have relevance to a number of physiological and ecological contexts. As these would be characterized by significant heterogeneity in pore sizes and geometries, further work will be necessary to translate the present results to those situations.

---

## [Author Response]

The following is the authors’ response to the previous reviews

**Public Reviews:**

**Reviewer #1 (Public review):**
The authors show experimentally that, in 2D, bacteria swim up a chemotactic gradient much more effectively when they are in the presence of lateral walls. Systematic experiments identify an optimum for chemotaxis for a channel width of ~8µm, a value close to the average radius of the circle trajectories of the unconfined bacteria in 2D. These chiral circles impose that the bacteria swim preferentially along the right-side wall, which indeed yields chemotaxis in the presence of a chemotactic gradient. These observations are backed by numerical simulations and a geometrical analysis.
**Reviewer #3 (Public review):**
This paper addresses, through experiment and simulation, the combined effects of bacterial circular swimming near no-slip surfaces and chemotaxis in simple linear gradients. The authors have constructed a microfluidic device in which a gradient of L-aspartate is established, to which bacteria respond while swimming while confined in channels of different widths. There is a clear effect that the chemotactic drift velocity reaches a maximum in channel widths of about 8 microns, similar in size to the circular orbits that would prevail in the absence of side walls. Numerical studies of simplified models confirm this connection.The experimental aspects of this study are well executed. The design of the microfluidic system is clever in that it allows a kind of "multiplexing" in which all the different channel widths are available to a given sample of bacteria.The authors have included a useful intuitive explanation of their results via a geometric model of the trajectories. In future work it would be interesting to analyze further the voluminous data on the trajectories of cells by formulating the mathematical problem in terms of a suitable Fokker-Planck equation for the probability distribution of swimming directions. In particular, this might help understand how incipient circular trajectories are interrupted by collisions with the walls and how this relates to enhanced chemotaxis.The authors argue that these findings may have relevance to a number of physiological and ecological contexts. As these would be characterized by significant heterogeneity in pore sizes and geometries, further work will be necessary to translate the present results to those situations.

Thanks to the referees' input and more work, we think our revised manuscript now meets the high standard of eLife

**Recommendations for the authors:**
The importance of the circular swimming chirality for the observed phenomenon could be further emphasized by actually using the word "chiral" or "chirality" in the text. Also indicating what would change is swimming were counterclockwise rather then clockwise would help the reader understand the key significance of chirality.

We thank the reviewer for this insightful suggestion. We agree that the chirality of the surface interaction is central to the observed phenomenon and should be explicitly highlighted to improve the reader's understanding.

In response, we have incorporated the terms "chiral" and "chirality" throughout the manuscript (Abstract, Introduction, Results, and Discussion) to emphasize this aspect. Furthermore, we have added a specific explanation in the Results section (the last paragraph of subsection “The cells in the right sidewall region dominated the chemotaxis of *E. coli* with lane confinements”) detailing the hypothetical scenario of counter-clockwise swimming. We clarify that in such a case, the hydrodynamic interaction would cause cells to veer left, resulting in up-gradient accumulation along the left sidewall rather than the right. We believe these additions significantly improve the clarity of the underlying physical mechanism.

**Reviewer #1 (Recommendations for the authors):**
I still have several comments that the authors may want to consider for the last version.- The run and tumble behavior of the cells at the surface remains puzzling and would need some more explanation in the text. Tumbles with no significant reorientation angle amount largely to smooth swimmers. How can a model based on run-and-tumbles be used to explain the difference between LSW and RSW?

We apologize for the lack of clarity regarding the surface run-and-tumble behavior. While it is true that surface tumbles often result in smaller reorientation angles compared to bulk swimming, they are not negligible and play a critical role in the observed asymmetry. As shown in the tumble angle distributions (Fig. 2E and 2F), the probability of a tumble angle exceeding π/2 is approximately 9% for sidewall trajectories and 30% for the middle area. This tumbling behavior leads to differences between the left sidewall (LSW) and right sidewall (RSW) in two key ways:

First, as detailed in our geometric analysis (Fig. 6), running cells following stable clockwise circular paths are geometrically favored to reach the RSW. Because cells moving up-gradient (towards the RSW) experience suppressed tumbling, they maintain these stable circular trajectories and accumulate effectively. Conversely, cells moving down-gradient (towards the LSW) experience enhanced tumbling. These frequent interruptions distort the circular trajectories required to reach the LSW, resulting in fewer bacteria entering the LSW compared to the RSW.

Second, once at the wall, the difference in tumbling frequency dictates retention. Majority of LSW cells are swimming down-gradient (LSW-DG) and thus tumble more frequently, increasing their probability of escaping the wall. Majority of RSW cells are swimming up-gradient (RSW-UG), suppressing tumbles and increasing their residence time at the wall.

The relevant clarifications have been included in the last paragraph of “Results” in the manuscript.

- Figure 5B would need more explanation. I still don't understand the different behaviors for the right and left side walls at small widths. Is it noise really or a more complex behavior? Since most of these calculations are based precisely on the shape of these curves it would be useful to discuss them in more detail.

We apologize for the lack of clarity. The behavior observed at small widths in Figure 5B is not noise; rather, it reflects the idealized nature of our simulation model.

In the simulation, bacteria were modeled as active particles without explicit steric exclusion for the flagella and cell body. Consequently, simulated cells retain the ability to reorient and turn freely even in very narrow lanes (*w* ≤ 6 μm), allowing the geometric sorting mechanism (which favors the RSW) to function efficiently even at small widths. This is why the simulation shows a distinct difference between LSW and RSW proportions in this regime.

In the experimental reality, however, the finite size of the bacterial body and flagella creates steric hindrance. In narrow channels, this physical constraint restricts the cells' ability to turn, thereby disrupting the circular swimming mechanism required to sort cells into the RSW. As a result, experimental data shows that the proportions of LSW and RSW cells tend to equalize in narrow channels (e.g., *w* = 6 μm in Fig. 4B), leading to a lower chemotactic drift velocity than predicted by the simulation.

We have added a discussion regarding these steric effects and the deviation at narrow widths to the Results section (the penultimate paragraph of subsection "Simulation of *E. coli* chemotaxis within lane confinement") in the revised manuscript.

- The importance of the chirality of the circular trajectories, although essential, remains insufficiently mentioned in the text.

We have incorporated the terms "chiral" and "chirality" throughout the manuscript (Abstract, Introduction, Results, and Discussion) to emphasize this aspect. Furthermore, we have added a specific explanation in the Results section (the last paragraph of subsection “The cells in the right sidewall region dominated the chemotaxis of *E. coli* with lane confinements”) detailing the hypothetical scenario of counter-clockwise swimming.

- It would be useful to color-code the trajectories of Figure 1B and alike with time.

Thank you for the suggestion. Now the trajectories in Fig. 1B have been redrawn. Distinct colors denote individual trajectories, with color intensity darkening to indicate time progression.